

# Genetic signatures of *Mycobacterium tuberculosis* Nonthaburi genotype revealed by whole genome analysis of isolates from tuberculous meningitis patients in Thailand

Olabisi Oluwabukola Coker[1], Angkana Chaiprasert[1], Chumpol Ngamphiw[2], Sissades Tongsima[2], Sanjib Mani Regmi[3], Taane G. Clark[4], Rick Twee Hee Ong[5], Yik-Ying Teo[5], Therdsak Prammananan[6] and Prasit Palittapongarnpim[7]

[1] Department of Microbiology, Faculty of Medicine Siriraj Hospital, Mahidol University, Bangkok, Thailand

[2] National Center for Genetic Engineering and Biotechnology, National Science and Technology Development Agency, Pathum Thani, Thailand

[3] Department of Microbiology, Gandaki Medical College, Pokhara Kaski, Nepal

[4] Faculty of Epidemiology and Population Health, Faculty of Infectious and Tropical Diseases, London School of Hygiene & Tropical Medicine, University of London, London, United Kingdom

[5] Saw Swee Hock School of Public Health, National University of Singapore, Singapore

[6] Tuberculosis Research Laboratory, Medical Molecular Biology Research Unit, National Center for Genetic Engineering and Biotechnology, National Science and Technology Development Agency, Pathum Thani, Thailand

[7] Department of Microbiology, Faculty of Science, Mahidol University, Bangkok, Thailand

Corresponding author
Angkana Chaiprasert,
angkana.cha@mahidol.ac.th

## ABSTRACT

Genome sequencing plays a key role in understanding the genetic diversity of *Mycobacterium tuberculosis (M.tb)*. The genotype-specific character of *M. tb* contributes to tuberculosis severity and emergence of drug resistance. Strains of *M. tb* complex can be classified into seven lineages. The Nonthaburi (NB) genotype, belonging to the Indo-Oceanic lineage (lineage 1), has a unique spoligotype and IS*6110*-RFLP pattern but has not previously undergone a detailed whole genome analysis. In addition, there is not much information available on the whole genome analysis of *M. tb* isolates from tuberculous meningitis (TBM) patients in public databases. Isolates CSF3053, 46-5069 and 43-13838 of NB genotype were obtained from the cerebrospinal fluids of TBM Thai patients in Siriraj Hospital, Bangkok. The whole genomes were subjected to high throughput sequencing. The sequence data of each isolate were assembled into draft genome. The sequences were also aligned to reference genome, to determine genomic variations. Single nucleotide polymorphisms (SNPs) were obtained and grouped according to the functions of the genes containing them. They were compared with SNPs from 1,601 genomes, representing the seven lineages of *M. tb* complex, to determine the uniqueness of NB genotype. Susceptibility to first-line, second-line and other antituberculosis drugs were determined and related to the SNPs previously reported in drug-resistant related genes. The assembled genomes have an average size of 4,364,461 bp, 4,154 genes, 48 RNAs and 64 pseudogenes. A 500 base pairs deletion, which includes *ppe50*, was found in all isolates. RD239, specific for members of Indo Oceanic lineage, and RD147c were identified. A total of 2,202 SNPs were common

to the isolates and used to classify the NB strains as members of sublineage 1.2.1. Compared with 1,601 genomes from the seven lineages of *M. tb* complex, mutation G2342203C was found novel to the isolates in this study. Three mutations (T28910C, C1180580T and C152178T) were found only in Thai NB isolates, including isolates from previous study. Although drug susceptibility tests indicated pan-susceptibility, non-synonymous SNPs previously reported to be associated with resistance to anti-tuberculous drugs; isoniazid, ethambutol, and ethionamide were identified in all the isolates. Non-synonymous SNPs were found in virulence genes such as the genes playing roles in apoptosis inhibition and phagosome arrest. We also report polymorphisms in essential genes, efflux pumps associated genes and genes with known epitopes. The analysis of the TBM isolates and the availability of the variations obtained will provide additional resources for global comparison of isolates from pulmonary tuberculosis and TBM. It will also contribute to the richness of genomic databases towards the prediction of antibiotic resistance, level of virulence and of origin of infection.

## INTRODUCTION

Tuberculosis (TB) remains a global threat despite efforts targeted towards its control. With recent advances in next generation sequencing, the analysis of bacterial whole genome sequences has contributed significantly to the understanding of virulence factors and antibiotic resistance of pathogenic bacteria (*Koser et al., 2013*; *Leopold et al., 2014*). Currently, there are software tools and databases that are used for predicting bacterial genotype, lineages and drug resistance profile from mycobacterial whole genome sequence data (*Benavente et al., 2015*; *Coll et al., 2015*). Availability of more whole genome data (processed and unprocessed), especially from genotypes not currently available, will contribute immensely to the profiling of pathogens.

Although tuberculosis is a curable disease, 9.0 million new cases and 1.5 million TB deaths were recorded in 2013 (*Zumla et al., 2015*). This is due in part to incomplete understanding of the variations that contribute to the pathogenesis and antibiotic resistance of *Mycobacterium tuberculosis*. There are two broad types of clinical TB disease; pulmonary (PTB) in which the site of infection is the lung and extra-pulmonary, including the more severe tuberculous meningitis (TBM), in which the bacteria cross the blood brain barrier to get into the cerebrospinal fluid (CSF) of the patient. The morbidity and mortality rate of TBM is higher than PTB (*Thwaites, van Toorn & Schoeman, 2013*). The genotype of the infecting mycobacterium has been shown to be one of the factors that contribute to the severity of the disease and can play a role in emergence of drug resistance, susceptibility to TBM, host response and in transmissibility (*Ford et al., 2013*; *Lopez et al., 2003*; *Nahid et al., 2010*; *Thwaites et al., 2008*). However the genetic factors that determine the association of different lineages of mycobacteria with different level of disease severity remain largely unknown.
There have been controversies in associating specific genotypes with morbidity or mortality from TB. A study in Thailand associated the modern Beijing genotype with a more severe disease progression when compared with other lineages (*Faksri et al., 2011*). However, in a study conducted in HIV patients in Vietnam, modern Beijing genotype had lower mortality rates than those infected with other lineages (*Tho et al., 2012*). Comparing strains isolated from TBM across genotypes on a whole genome scale may provide better understanding of factors that contribute to the severity of the disease.

IS*6110* based restriction fragment length polymorphism (RFLP) is an internationally recognized method for genotyping mycobacteria (*Thierry et al., 1990*; *van Embden et al., 1993*). Nonthaburi strains of *M. tuberculosis* were first identified in Thailand by its IS*6110*-RFLP patterns, usually containing 9-14 bands. Subsequent spoligotyping revealed that the Nonthaburi type has a spoligotype octal code 674000003413771 specifying the East-Asian India 2 Nonthaburi (EAI2-Nonthaburi) genotype (*Palittapongarnpim et al., 1997*). It has been reported in lower percentages from many countries such as the Netherlands, Australia, USA, Sweden, Saudi Arabia, Tunisia, and Taiwan. However, the origin of the isolates is likely to be South East Asia, as more isolates are from countries such as Indonesia, Laos PDR, Vietnam, Cambodia, Philippines and Thailand (*Demay et al., 2012*).

As of this date, only relatively little information is available on the genetic characteristics of the Nonthaburi strains. Three Nonthaburi strains were isolated from the CSF samples of TBM patients at Siriraj Hospital, Mahidol University, Thailand. For a deeper understanding of the characteristics of these isolates, genome-wide scale analysis and drug susceptibility pattern to anti-tuberculosis drugs were performed and compared to the reference strain *M. tuberculosis* H37Rv (NC_000962.3). The single nucleotide polymorphism (SNPs) common to the isolates were compared with SNPs from 1,601 genomes from the 7 different lineages and various sublineages of *M. tuberculosis* complex (MTBC). The whole genome sequence of the isolates were assembled into draft genomes, annotated and have been deposited into NCBI database for public access. Prior to our study, there was no complete or draft genome belonging to the Nonthaburi genotype of *M. tuberculosis* in the database.

## METHODS

### Selection of strains

Three isolates, CSF3053, 46-5069 and 43-13838, identified to belong to Nonthaburi genotype by IS*6110*-RFLP, were selected from the stock of samples collected from the CSF of TBM patients at the Drug Resistant Tuberculosis Research Fund Laboratory, Department of Microbiology, Faculty of Medicine Siriraj Hospital, Mahidol University, Thailand.

### Genomic DNA extraction

Stock culture of selected strains, stored at −70 °C in MH79 broth containing 15% glycerol, were subcultured on Loewenstein-Jensen medium and incubated for 4 weeks at 37 °C. DNA extraction was carried out using cetyltrimethylammonium bromide (CTAB)-lysozyme enzymatic method as earlier described (*Larsen et al., 2007*).

## Spoligotyping

Spacer oligonucleotide typing, a polymerase chain reaction (PCR) based method used in typing *M. tuberculosis* was performed following the methods earlier described (*Gori et al., 2005*).

## Whole genome sequencing and analysis

Genomic DNA samples isolated from the three isolates were sequenced at Macrogen Inc., Seoul, South Korea on the HiSeq 2000 platform with insert size of 300 bp (Illumina, San Diego, CA, USA) yielding 100 bp paired end reads. The qualities of the sequences were assessed with FastQC software (www.bioinformatics.babraham.ac.uk/projects/fastqc) to determine the parameters used for trimming. Bases with quality of less than 5, reads with average of quality less than 20 for every four bases, and reads with lengths that are less than 45 bases were discarded using Trimmomatic software (*Bolger, Lohse & Usadel, 2014*) (version 0.33). The trimmed sequences were aligned to the reference strain *M. tuberculosis* H37Rv (NC_000962.3) using the short reads aligner, Bowtie2 (version 2.2.0) (*Langmead & Salzberg, 2012*). The genomic coverage was estimated using Bedtools (version 2.18) (*Quinlan & Hall, 2010*). The fold coverage is estimated as the number of reads supporting a particular nucleotide position on the genome. Variant calling was performed on the aligned sequences using the Genome Analysis Tool Kit (GATK) (version 3.3) haplotype caller (*McKenna et al., 2010*) with minimum calling confidence threshold set at phred score 30. Point allelic variation at any position within the genome when compared with the reference H37Rv genome (NC_000962.3) is considered a single nucleotide polymorphism (SNP).

SnpEFF (*Cingolani et al., 2012*) (version 4.0) software was used to annotate the SNPs. The SNPs were filtered using standard hard filtering parameters according to GATK Best Practices Recommendations (*DePristo et al., 2011*; *Van der Auwera et al., 2013*). Variants with QualByDepth <2.0, FisherStrand >60, RMSMapping quality <40, MappingQualityRankSumTest <−12.5 and ReadPosRankSumTest <−8 were filtered. All SNPs were confirmed using Integrated Genomic Viewer (IGV) (*James et al., 2011*) (version 2.0). The SNPs were further grouped according to the functions of the genes in which they were found in the genome when compared to the reference genome H37Rv (NC_000962.3). We evaluated SNPs in groups of genes considered to be essential, drug resistance related, virulence related, contain known epitopes and associated with efflux pumps.

The Whole Genome Shotgun project has been deposited at DDBJ/EMBL/GenBank under the accession numbers LGCH00000000, LGCG00000000 and LGCF00000000. The versions described in this paper are LGCH01000000, LGCG01000000 and LGCF01000000 for CSF3053, 46-5069 and 43-13838 respectively. The raw sequences have been deposited to the short read archive (SRA) of NCBI under accession numbers SRX1094547, SRX1094546 and SRX1094545 for isolates CSF3053, 46-5069 and 43-13838 respectively.

## Determination of principle genetic group, lineage and sequence type

Nucleotide alleles at positions 7585 and 2154724 were investigated to determine the principal genetic group of the isolates as earlier defined (*Sreevatsan et al., 1997*). To

determine the lineage of the isolates, SNPs specific to different lineages as earlier reported (*Coll et al., 2014b*) were investigated.

## Draft genome assembly

The paired-end raw reads of the isolates were assembled into draft genomes by using the *de novo* assembly algorithm of CLC Genomics Workbench (version 7.5) which works by using a de Bruijn graph (http://www.clcbio.com/). The minimum contig output was set at 200 bp long. Annotation of the draft genome was performed by Rapid Annotation using Subsystem Technology (RAST) (http://www.nmpdr.org/) and by NCBI Prokaryotic Genome Annotation Pipeline (PGAP) (http://www.ncbi.nlm.nih.gov/genome/annotation_prok/).

## Comparison of Nonthaburi isolates with isolates from other lineages

The SNPs that are common to the three isolates were compared with 92,000 SNPs from 1,601 genomes of MTBC previously reported (*Coll et al., 2014a*) (http://pathogenseq.lshtm.ac.uk/phytblive/index.php). These include 121, 390, 189, 856, 17, 11, and 6 genomes from lineages 1, 2, 3, 4, 5, 6, and 7 respectively. Eleven samples from *M. bovis* were also included.

## Large sequence polymorphism determination

Regions of differences when compared with reference strain H37Rv (NC_000926.3) were determined by using the indel and structural variants determination tool of CLC Genomics Workbench (version 7.5) (http://www.clcbio.com) and Bedtools (version 2.18) (*Quinlan & Hall, 2010*). The regions of deletions were confirmed with PCR using primers CF (CATCCGCACCGAACCTGTAA) and CR (AACCGTTCACGACAAGCAAC), AF (GCCCAACCTGATTGGTTTCG) and AR (CAAACGCTCGCCATGATCTC), BF (TCGACTGCCATACAACCTGC) and BR (ACTTCCGGTGGTAACAGTGC) respectively for RD239, RD147c and newly identified deletion of 500 bp between 3501224-3501724 (*M. tuberculosis* H37Rv (NC_000962.3 genome numbering). The reactions were performed with initial denaturation at 94 °C and 30 cycles of denaturation for 1 min, annealing of primers at 60 °C for 1 min and extension with platinum *Taq* DNA polymerase for 1 min at 68 °C. Final extensions were performed at 68 °C for 10 min. The reactions were performed as recommended by the manufacturer of the DNA polymerase.

## Drug susceptibility testing

The susceptibility of the isolates to first line drugs and other second-line anti-tuberculosis drugs was investigated using the standard agar proportion method (*Larsen et al., 2007*). The drug concentrations used in the test comprise 0.2 mg/l isoniazid, 1.0 mg/l rifampicin, 2.0 mg/l streptomycin, 5.0 mg/l ethambutol, 1.0 mg/l linezolid, 6.0 mg/l amikacin, 5.0 mg/l ethionamide, 2.0 mg/l paraaminosalycic acid, 2.0 mg/ml ofloxacin, 2.0 mg/l moxifloxacin, 2.0 mg/l gatifloxacin, 1.0 mg/ml sitafloxacin, 6.0 mg/l kanamycin, 2.0 mg/l ciprofloxacin, 2.0 mg/l levofloxacin, and 3.0 mg/l clarithromycin. Growth equal to or more than 1% on drug containing media compared to drug free media was recorded as drug resistance. The phenotypic drug testing was performed on the initial isolates from the patients and repeated on the stock cultures.

### Ethical approval

The study was approved by the Institutional review board (IRB) of Faculty of Medicine Siriraj Hospital, Mahidol University SiEC No. 152/2549.

## RESULTS AND DISCUSSION

For the three isolates CSF3053, 46-5069 and 43-13838, an average of 99.1% of raw reads mapped to the reference genome. On the average, 99.8% of the reference was covered to at least 1-fold coverage. The depth across all the positions covered by the reads was about 1,056-fold on the average (Table 1).

### Genome assembly

The sequences of the isolates were assembled and annotated as described in Methods. 159 contigs with $N_{50}$ of 69,028, 173 contigs with $N_{50}$ of 63,852, and 177 contigs with $N_{50}$ of 63,019 contigs were obtained for CSF3053, 43-5069 and 46-13838 respectively. All isolates have 65.5 % guanine/cytosine (GC) content, typical of mycobacteria. The draft genomes have an average size of 4,364,461 bp, 4,154 genes, 48 rRNAs and 64 pseudogenes. Details of the assembly and annotation are shown in Table 1.

### Single nucleotide polymorphisms

Point allelic variations at any position within the genome when compared with the reference H37Rv genome (NC_000962.3) were investigated.

In total, 2,202 positions were found to have similar allelic changes (SNPs) in all isolates as shown in Fig. 1. 1,963 are in coding regions (754 synonymous, 1209 (61.6%) non synonymous) and 239 are intergenic. In this study, CSF3053, 46-5069 and 43-13838 have 10, 7 and 49 unique SNPs respectively. 43-13838 and CSF3053 have 23 SNPs in common, CSF3053 and 46-5069 have 99 SNPs in common, while 43-13838 and 46-5069 have 7 SNPs in common. Using the SNPs, the isolates were found to belong to lineage 1 with the presence of allele C/A and G/C at positions 2154724 and 7585 resulting in *katG* R463L and *gyrA* S95T respectively (*Sreevatsan et al., 1997*). Using a recently developed SNP barcode (*Coll et al., 2014a*), the isolates were found to be specific to Indo Oceanic lineage 1.2.1, with nucleotide changes G/A at position 615938, C/A at position 3479545, G/C at position 4244420 and G/C at position 9260.

The 2,202 SNPs that were found to be common to the isolates in this study were compared with 92,000 SNPs from 1,601 genomes of MTBC that were analyzed previously. These include 121, 390, 189, 856, 17, 11, and 6 genomes from lineages 1, 2, 3, 4, 5, 6, and 7, respectively. Eleven samples from *M. bovis* were also included (*Coll et al., 2014a*). The common SNPs were used to position the strains on a phylogenetic tree compared to other strains and lineages of MTBC as shown in Fig. S1. Nucleotide change G/C at position 2342203 was found only in the isolates in this study when compared with the 1,601 MTBC genomes. There is evidence from macrophage systems that strain-to-strain variability affects phenotypic outcomes (*McEvoy et al., 2012*). Phylogeographic strain variation may therefore have considerable effect on the development of new diagnostic tools, vaccines and drugs.

Coker et al. (2016), *PeerJ*, DOI 10.7717/peerj.1905

**Table 1  Statistics of whole genome sequencing, genome assembly and annotation.** Gross statistics of the whole genome sequence data, mapping of reads, assembly of draft genome and annotation for isolates CSF-3053, 46-5069 and 43-13838. Length of reference genome (*M. tuberculosis* H37Rv, NC_000962.3) is 4,411,532 base pairs.

| Isolate | Total reads | % of reads mapped to reference | % of Reference covered | Number of contigs | N50 | Fold coverage of positions in the genome | GC content (%) | Number of predicted Genes | No. of predicted RNA genes | No. of predicted pseudo genes |
|---|---|---|---|---|---|---|---|---|---|---|
| CSF-3053 | 50,004,564 | 99.96 | 99.78 | 159 | 69,028 | 1329.0 | 65.5 | 4153 | 48 | 62 |
| 46-5069 | 44,478,206 | 98.67 | 99.82 | 173 | 63,852 | 920 | 65.5 | 4159 | 48 | 63 |
| 43-13838 | 40,767,970 | 98.69 | 99.80 | 177 | 63,019 | 920 | 65.5 | 4150 | 48 | 67 |

**Notes.**

GC, guanine/cytocine.
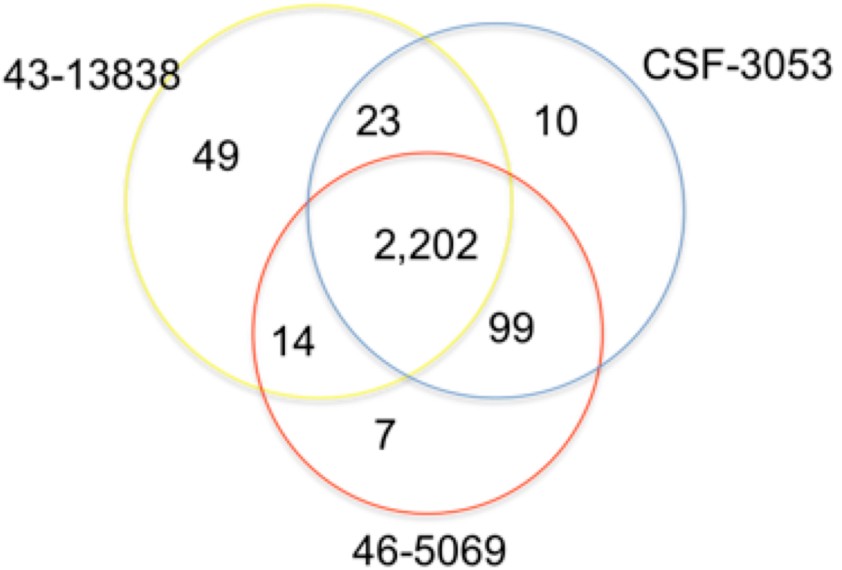

**Figure 1 Distribution of single nucleotide polymorphisms in isolates CSF3053, 46-5069 and 43-13838.** Venn diagram showing the distribution of the single nucleotide polymorphisms (SNPs) observed in isolates CSF-3053 (blue), 46-5069 (red) and 43-13838 (yellow). CSF3053, 46-5069 and 43-13838 have 10, 7 and 49 unique SNPs respectively. 43-13838 and CSF3053 have 23 SNPs in common, CSF3053 and 46-5069 have 99 SNPs in common, while 43-13838 and 46-5069 have 14 SNPs in common. 2,202 SNPs are common to all isolates.

SNP C/T at position 3378828 was reported to be unique to members of lineage 1 (*Coll et al., 2014a*). Although this SNP was found in many genomes belonging to lineage 1, we found out that it was absent in the three isolates in this study and in 6 other Nonthaburi isolates from Thailand and the Netherlands used in previous studies which are grouped under lineage 1. This indicates that the allele change at this position may be specific only to a sub-branch of lineage 1. Synonymous SNP T/C at position 28910, non-synonymous SNP C/T at position 152178 resulting in Thr344Ile in *pepA* gene and intergenic SNP C/T at position 1180580 were found only in Nonthaburi isolates from Thailand. They were not found in any genome belonging to lineages 2, 3, 4, 5, 6 and 7. Within lineage 1, these SNPs were found only in Thai Nonthaburi isolates, from previous study (*Coll et al., 2014a*), and the isolates in this study. They were however absent in the Nonthaburi genotype isolates from the Netherlands. *pepA* gene is a probable serine protease with the exact function unknown. It is in the intermediary metabolism and respiration functional category. Its mRNA was found to be upregulated after 96 h of starvation (*Betts et al., 2002*), suggesting its role in the adaptation of mycobacteria to extreme conditions. The association of the SNPs at these positions with Thailand warrants further investigation.

## Large sequence polymorphism

Region of difference RD239 that is specific to lineage 1 of MTBC and previously reported RD147c, not specific to lineage 1, were found in all the three isolates. In addition, a region of deletion of 500 bp between 3501224-3501724 (*M. tuberculosis* H37Rv (NC_000962.3 genome numbering) comprising Rv3135 (*ppe50*), was observed in all isolates. The details
**Table 2  Regions of deletion and affected open reading frames found in isolates CSF-3053, 46-5069 and 43-13838.** All regions were confirmed by PCR reaction as described in methods.

| Region in reference genome (H37Rv, NC_000962.3) | Length | Region of difference | Open reading frame (ORF) affected |
| --- | --- | --- | --- |
| 1718912–1721213 | 2302 | RD147c | *Rv1526c* |
| | | | *Rv1525 (wbbL2)* |
| | | | *Rv1526c* |
| 3501225–3501723 | 499 | This study | *Rv3135* |
| 4092082–4092921 | 840 | RD239 | *Rv3651* |

of the deletions as well as the affected open reading frames are shown in Table 2. The deletions were confirmed with PCR (see Figs. S2, S3 and S4). The PE-PPE protein class, while not well characterized, represents the third most abundant category of mycobacterial proteins and showed the most consistent expression during infection (*Kruh et al., 2010*). Although PPE50 has a yet unknown function, it was listed among promising therapeutic target in tuberculosis treatment based on its expression, and homology to human and other microbial proteins (*Raman, Yeturu & Chandra, 2008*). The deletion of this gene may be a means of evading recognition by the host immune system.

Deletions have been shown to have a wide range of effects on *M. tuberculosis* including association with an increased probability of transmission (*Tsolaki et al., 2004*).

## Polymorphisms in drug resistance associated genes

Despite being isolated from patients with severe form of tuberculosis, drug susceptibility tests results show that the three isolates are susceptible to first line drugs; isoniazid, rifampicin, ethambutol and streptomycin, and to quinolones: ciprofloxacin, ofloxacin, gatifloxacin, moxifloxacin, levofloxacin, and sitafloxacin. They were also found to be susceptible to linezolid, amikacin, ethionamide, paraaminosalicylic acid, kanamycin and clarithromycin.

However, 37 SNPs were found in drug-resistant related genes reported in TBdream database and other earlier published reports (*Sandgren et al., 2009*). Nineteen are synonymous while 18 are non synonymous. Non synonymous mutations Gly312Ser of *kasA* gene and Ile73Thr in *efpA* were previously reported to be associated with isoniazid resistance (*Mdluli et al., 1998*; *Ramaswamy et al., 2003*), but were found in our isolates. Association between these mutations and resistance to isoniazid needs to be confirmed. *iniA* gene and Rv1592c were reported to be associated with tolerance to isoniazid (*Colangeli et al., 2005*; *Ramaswamy et al., 2003*). In our analysis, mutations His481Gln in *iniA* gene and Ile322Val in *Rv1592c* were found. These positions may not be associated with the supposed roles of these genes in isoniazid resistance.

Polymorphism exists at position 237 of *nudC* in *M. tuberculosis* isolates (*Wang et al., 2011*). In particular, the amino acid change Gln237Pro in *nudC* is found in the Indo Oceanic and West African lineages. It was demonstrated to prevent dimer formation and results in the loss of activity of the enzyme. It was also shown to degrade the active forms of isoniazid and ethionamide (*Wang et al., 2011*). We however found this codon change in

all isolates in this study. This suggests the non-involvement of the amino acid change at this position in resistance to both drugs.

Mutations Cys110Tyr in *embR*, Thr270Ile and Asn394Asp in *embC*, Pro913Ser in *embA* and Glu378Ala in *embB*, were previously reported to be involved in ethambutol resistance (*Ramaswamy et al., 2000*; *Srivastava et al., 2009*). However, these mutations were found in this study. Mutation Ser257Pro in *rmlD* was suspected to be involved in isoniazid and ethambutol resistance (*Ramaswamy et al., 2000*). This was however found in all isolates considered in this study. Mutations Glu21Gln in *gyrA*, Ile322Val in Rv1592c, Arg463Leu in *katG*, and Arg93Leu in *cycA* were found to be common to the isolates in this study. They have also been reported to be common to pan-susceptible and drug-resistant *M. tuberculosis* sequence type 10 Beijing isolates (*Regmi et al., 2015*). Our results confirm that these mutations are polymorphic rather than being involved in drug resistance. The details of the synonymous and non-synonymous SNPs found in drug-resistant related genes and the predicted protein variation effects are shown in Table 3.

## Polymorphisms in virulence genes, efflux pump related genes, and essential genes

Oftentimes, mutations provide selective advantage to an organism in a particular environment. Some non-synonymous mutations in *rpoC* gene have been shown to result in higher competitiveness *in vitro* and have higher fitness *in vivo* evidenced by their prevalence across patient populations (*Comas et al., 2012*). In this study, we found Ala172Val mutation in *rpoC* gene in all isolates.

We also sought to determine polymorphisms in genes that play important roles in the survival and pathogenesis of *M. tuberculosis*. Of particular interest are the genes that are involved in the evasion of the host immune system. SNPs in 37 mycobacteria virulence related genes were found to be common to the isolates. Twenty nine of the SNPs are non-synonymous. Polyketide synthases (PKs) are group of genes involved in the synthesis of polyketides which are structurally complex compounds produced by organisms for survival advantage. Some mycobacteria PKs genes such as *pks15, pks1, pks10, pks12, pks5*, and *pks7* are known to be involved in virulence (*Reed et al., 2004*; *Rousseau et al., 2003*; *Sirakova et al., 2003*; *Tsenova et al., 2005*). Insertion of 7 base pairs was found in *pks15/1* junction in all isolates. The presence of the 7 base pair insertion leads to a frame shift that results in the loss of stop codon of *pks15*. This results in a continuous transcription of *pks15* and *pks1*. This was previously associated with the more virulent phenotype of the modern Beijing family, but such claim has since been refuted as it can be found across the seven lineages. The implication of the insertion needs further experiments to understand. Two mutations Ile474Met and Thr604Ala were found in *nuoG* gene. *nuoG* is a probable NADH dehydrogenase, reported to be involved in apoptosis inhibition (*Velmurugan et al., 2007*). Mutation Arg463Leu was found in *katG*, a gene previously implicated in inhibiting antimicrobial effectors of the macrophage (*Ng et al., 2004*). Protein kinases such as *pknD* and *pknG* are important virulent factors of *M. tuberculosis*. *pknD* has been reported to play a role in the infection of the host's central nervous system by *M. tuberculosis* (*Be, Bishai & Jain, 2012*; *Cowley et al., 2004*). Gln472Pro mutation in *pknD* was found in all isolates. *virS*

**Table 3** **Common SNPs found in drug resistance related genes in isolates CSF-3053, 46-5069 and 43-13838.** The reference genome positions, nucleotide change, amino acid change and effect of single nucleotide polymorphisms in drug resistance related genes that are common to isolates CSF3053, 46-5069 and 43-13838. The protein variation was determined by Protein Variation Effect Analyzer (PROVEAN), a web based protein variation analysis tool (*Choi et al., 2012*).

| Position in reference genome (H37Rv, NC_000962.3) | Nucleotide change | Amino acid change | Protein variation effect | Gene | Associated drug | References |
|---|---|---|---|---|---|---|
| 6112 | G>C | Met291Ile | Deleterious | *gyrB* | Quinolones | *Guillemin, Jarlier & Cambau (1998)* |
| 7362 | G>C | Glu21Gln | Neutral | *gyrA* | Quinolones | *Guillemin, Jarlier & Cambau (1998)* |
| 7585 | G>C | Ser95Thr | Neutral | *gyrA* | Quinolones | *Guillemin, Jarlier & Cambau (1998)* and *Kapur et al. (1995)* |
| 8452 | C>T | Ala384Val | Deleterious | *gyrA* | Quinolones | *Guillemin, Jarlier & Cambau (1998)* |
| 9143 | T>C | Ile614Ile | | *gyrA* | Quinolones | *Guillemin, Jarlier & Cambau (1998)* |
| 9260 | G>C | Leu653Leu | | *gyrA* | Quinolones | *Guillemin, Jarlier & Cambau (1998)* |
| 9304 | G>A | Gly668Asp (N) | Neutral | *gyrA* | Quinolones | *Guillemin, Jarlier & Cambau (1998)* |
| 412280 | T>G | His481Gln | Neutral | *iniA* | Ethambutol | *Ramaswamy et al. (2003)* |
| 575368 | T>C | Asp7Asp | | *Rv0486* | Isoniazid/Ethionamide | *Projahn et al. (2011)* |
| 763031 | T>C | Ala1081Ala | | *rpoB* | Rifampicin | *Taniguchi et al. (1996)* |
| 763531 | G>C | Pro54Pro | | *rpoC* | Rifampicin | *Comas et al. (2012)* |
| 763884 | C>T | Ala172Val | Neutral | *rpoC* | Rifampicin | *Comas et al. (2012)* |
| 763886 | C>A | Arg173Arg | | *rpoC* | Rifampicin | *Comas et al. (2012)* |
| 1406312 | A>G | His343His | | *Rv1258c* | Streptomycin | *Siddiqi et al. (2004)* |
| 1417019 | C>T | Cys110Tyr | Deleterious | *embR* | Ethambutol | *Ramaswamy et al. (2000)* |
| 1674162 | C>T | Gly241Gly | | *fabG1* | Isoniazid | *Lavender et al. (2005)* |
| 1792777 | T>C | Ile322Val | Neutral | *Rv1592c* | Isoniazid | *Ramaswamy et al. (2003)* |
| 1792778 | T>C | Glu321Glu | | *Rv1592c* | Isoniazid | *Ramaswamy et al. (2003)* |
| 2154724 | C>A | Arg463Leu | Neutral | *katG* | Isoniazid | *Heym et al. (1995)* |
| 2518132 | C>T | Thr6Thr | | *kasA* | Isoniazid | *Lee et al. (1999)* |
| 2519048 | G>A | Gly312Ser | Neutral | *kasA* | Isoniazid | *Lee et al. (1999)* |
| 2521342 | T>C | Asp200Asp | | *accD6* | Isoniazid | *Ramaswamy et al. (2003)* |
| 3154414 | A>G | Ile73Thr | Neutral | *efpA* | Isoniazid | *Ramaswamy et al. (2003)* |
| 3571834 | T>G | Gln237Pro | Neutral | *nudC* | Isoniazid/Ethionamide | *Wang et al. (2011)* |
| 3647041 | A>G | Ser257Pro | Neutral | *rmlD* | Ethambutol | *Ramaswamy et al. (2000)* |
| 3647591 | A>G | Asn73Asn | | *rmlD* | Ethambutol | *Ramaswamy et al. (2000)* |
| 4049254 | G>A | Leu243Leu | | *folP1* | Para-aminosalicylic acid | *Mathys et al. (2009)* |
| 4240671 | C>T | Thr270Ile | Neutral | *embC* | Ethambutol | *Ramaswamy et al. (2000)* |
| 4241042 | A>G | Asn394Asp | Deleterious | *embC* | Ethambutol | *Ramaswamy et al. (2000)* |
| 4242643 | C>T | Arg927Arg | | *embC* | Ethambutol | *Ramaswamy et al. (2000)* |
| 4243580 | G>A | Val116Val | | *embA* | Ethambutol | *Telenti et al. (1997)* |
| 4244420 | G>C | Val396Val | | *embA* | Ethambutol | *Telenti et al. (1997)* |
| 4245969 | C>T | Pro913Ser | Deleterious | *embA* | Ethambutol | *Ramaswamy et al. (2000)* and *Telenti et al. (1997)* |
| 4247578 | G>A | Leu355Leu | | *embB* | Ethambutol | *Telenti et al. (1997)* |
| 4247646 | A>C | Glu378Ala | Neutral | *embB* | Ethambutol | *Telenti et al. (1997)* |
| 4407588 | T>C | Ala205Ala | | *rsmG* | Streptomycin | *Okamoto et al. (2007)* |
| 4407873 | C>A | Val110Val | | *rsmG* | Streptomycin | *Okamoto et al. (2007)* |

is a transcription regulator that belongs to AraC family. Its attenuation in a mouse model resulted in an increased animal survival (*Gupta, Jain & Tyagi, 1999*; *Singh et al., 2003*). We found mutation Leu316Arg in this gene in all isolates.

Stop codon was gained after Arg305 in *PStA1*, an inorganic phosphate ABC transporter. Stop codon was however lost in *Rv1504*. The stop codon was replaced with glutamine as codon 200. *Rv1504* and *PstA1* were reported to be involved in the adaptation and survival of mycobacteria in macrophages (*Brodin et al., 2010*; *Rengarajan, Bloom & Rubin, 2005*).

Non-synonymous and synonymous SNPs were found in other genes involved in various other functions related to virulence such as synthesis of complex and simple lipids, cell wall proteins, lipoproteins, cholesterol metabolism, secretion systems, protein kinases, metal transporter proteins, two component systems and other proteins of unknown functions (Table S1).

Efflux pumps play roles in drug resistance, cell physiology, detoxification and virulence of *M. tuberculosis* (*Nikaido, 2009*). Ten synonymous SNPs and 15 non-synonymous SNPs were found in efflux pump related genes. One stop codon was gained by *Rv2994*, a predicted transmembrane protein involved in efflux system (Table S2).

Twenty eight SNPs were observed in genes with known epitopes, 11 are synonymous while 17 are non-synonymous (Table S3).

In addition, 316 SNPs were found in essential genes, 135 are synonymous, 181 are non-synonymous. A start codon was lost in *pabB* gene. *pabB* is a cell membrane associated gene that encodes *para*-aminobenzoate synthetase component-I involved in the biosynthesis of *p*-aminobenzoate, a precursor of folate biosynthesis (*Sassetti, Boyd & Rubin, 2003*; *Zheng et al., 2008*). The details of the position, nucleotide change, amino acid change and the genes involved are presented in Table S4.

The association of the SNPs or deletions reported in this study to TBM needs further investigations. This can be done by comparing them with variations from PTB cases, to determine exclusive associations with TBM. Furthermore, The involvement of the reported allelic changes in the functions of the various genes from which they were found can be verified by site directed mutagenesis in laboratory strains of *M. tuberculosis*, and subsequent animal experiments.

## CONCLUSION

Genetic factors that contribute to the ability of infecting mycobacteria in causing TBM remain largely unknown. We have presented a detailed analysis of the polymorphism existing in the genome of Nonthaburi isolates from TBM patients, when compared to reference strain *M. tuberculosis* H37Rv (NC_000962.3). The polymorphisms were compared to 1,601 genomes representing the members of the 7 MTBC lineages. Uniqueness of certain SNPs to certain genotypes, countries or region such as found in this study may be useful epidemiologically to determine the origin of an infection and potential level of disease severity. We have also presented the first draft genomes of *M. tuberculosis* Nonthaburi genotype.

Many studies have reported the SNPs playing roles in drug resistance in many drug-resistant related genes. These have majorly formed the basis for the development of some

databases. It is equally important to report polymorphisms found in these genes from drug susceptible strains so that SNPs that are not involved in resistance to drugs but present in the drug resistance related genes could be filtered out in the process of predicting drug resistance. Our results will also form a basis for comparison with other genotypes of mycobacteria isolated from the CSF of TBM or sputum of PTB patients in order to identify potential factors contributing to TBM.

### Funding

The financial support for the study are from Mahidol University as postdoctoral scholarship to BOC via AC and some material support from JST/NSTDA grant No. P-12-01777. The funders had no role in study design, data collection and analysis, decision to publish, or preparation of the manuscript.

### Grant Disclosures

The following grant information was disclosed by the authors:
Mahidol University as postdoctoral scholarship.
JST/NSTDA: P-12-01777.

### Competing Interests

The authors declare there are no competing interests.

### Author Contributions

- Olabisi Oluwabukola Coker conceived and designed the experiments, performed the experiments, analyzed the data, wrote the paper, prepared figures and/or tables, reviewed drafts of the paper.
- Angkana Chaiprasert conceived and designed the experiments, analyzed the data, contributed reagents/materials/analysis tools, wrote the paper, reviewed drafts of the paper.
- Chumpol Ngamphiw performed the experiments, analyzed the data, reviewed drafts of the paper.
- Sissades Tongsima analyzed the data, reviewed drafts of the paper.
- Sanjib Mani Regmi performed the experiments, reviewed drafts of the paper.
- Taane G. Clark reviewed drafts of the paper, suggested the position of NB in phylogenetic tree.
- Rick Twee Hee Ong, Yik-Ying Teo and Therdsak Prammananan contributed reagents/materials/analysis tools, reviewed drafts of the paper.
- Prasit Palittapongarnpim reviewed drafts of the paper.

### Ethics

The following information was supplied relating to ethical approvals (i.e., approving body and any reference numbers):

InstitutionL Review Board (IRB) of Faculty of Medicine Siriraj Hospital, Mahidol University SiEC No. 152/2549.

## DNA Deposition

The following information was supplied regarding the deposition of DNA sequences:

DDBJ/EMBL/GenBank with accession numbers LGCH01000000, LGCG01000000 and LGCF01000000 for CSF3053, 46-5069 and 43-13838 respectively. The raw sequences have been deposited to the short read archive (SRA) of NCBI under accession numbers SRX1094547, SRX1094546 and SRX1094545 for isolates CSF3053, 46-5069 and 43-13838 respectively.

## Data Availability

The raw sequences have been deposited to the short read archive (SRA) of NCBI under accession numbers SRX1094547, SRX1094546 and SRX1094545 for isolates CSF3053, 46-5069 and 43-13838 respectively.

## Supplemental Information

Supplemental information for this article can be found online at http://dx.doi.org/10.7717/peerj.1905#supplemental-information.

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
