# Peer review of "Genetic signatures of Mycobacterium tuberculosis Nonthaburi genotype revealed by whole genome analysis of isolates from tuberculous meningitis patients in Thailand"

_PeerJ, doi:10.7717/peerj.1905_

## Round 0.1 · original submission · Major Revisions

Four independent reviewers have evaluated your manuscript and identified some aspects that would require your attention before it is ready for publication. There is a consensus among the reviewers on a need of a phylogenetic tree. Furthermore, the manuscript would benefit from the inclusion of additional methodological details to ensure reproducibility as pointed out by the reviewers.

Reviewer 1 ·

Basic reporting

1) I was unable to acces raw data cause provided SRA experiments are no public.

2) Reference for SnpEFF is missing (line 168)

Experimental design

I have a few comments/recomendations about the analysis you used.

1)I'm not an expert in M tuberculosis H37Rv, but I know that this organism is a partial diploid system (it is called merodiploid). It has more that one copy of different genes such as Tat (TatAC and TatB) or ideR. According with you methods, you have used GATK with the haplotype caller. So my question is, wich type of ploid parameter have you used ?.

2)I've used CLC bio for a couple of times and I have compared results against Soapdenovo2, velvet and MIRA. CLC was far to be the best. So I think that the quality of your results should improve this way (just a recommendation)

Validity of the findings

No Comments

Additional comments

According to PeerJ policies, the raw data should be accesible for reviewers, but I could not find them and SRA ids are no public

·

Basic reporting

Data is shared on SRA and in Genbank.

Experimental design

Some part of the methods section needs to be better described to be able to be reproduced.

Validity of the findings

I think the manuscript would benefit from a phylogenetic tree that visualizes the phylogenetic position of these strains compared to other MTB strains. Additionally it would be a great addition to investigate the important mutations using SIFT and to identify the origin of the additional 500k reads not mapped to the reference genome. See General Comments for the Author for details.

Additional comments

Coker et al., presents sequence data, assembly and analysis of three M. tuberculosis Nonthaburi isolates from tuberculous meningitis patients in Thailand. The genomes of the three strains were sequenced to average 1000X and assembled using both the reference genome (H37Rv) and by de novo assembly. SNPs are determined from the alignment based approaches and larger deletions using the de novo assembly. The genomic variation is then used to investigate lineage, drug resistance and polymorphisms in virulence genes. The drug resistance SNPs are compared to drug susceptibility tests and the differences compared to previous knowledge discussed.

In general the manuscript reads well, however it could benefit from a phylogenetic tree visualizing the relationship of the three Nonthaburi strains in relation to other MTB strains. This could be done using the data from Coll et al., 2014a or similar.

The important mutations could be assessed using SIFT for deleteriousness – this would be particular interesting for the mutations previously associated to resistance but where these strains are susceptible. This could be added to eg. Table 3 and the supplementary tables 6-8.

Why do the “46-5069” and “43-13838” strains only have 98.7% reads mapped to the reference genome opposed to the “CSF-3053” strain having 99.96% of reads mapped? All three strains cover the same amount of the reference genome (99.78%-99.82%) indicating that this is extra sequence/contamination? The un-mapped reads correspond to around 500k reads from each of the strains – quite a substantial amount. It would be interesting to map these reads to e.g. all bacteria or the MTB complex. Additionally, a figure showing the depth distributions of the de novo assembled contigs/scaffolds would help indicate whether this is contamination (much lower depth) or additional MTB sequence in these two strains (similar depth to the genome).

Minor comments:
- Please report the insert size of the paired end libraries, this is useful for assessing de novo assembly statistics
- Please define the filters used for SNP filtration, currently it states “default” which does not make it possible to reproduce the analysis.
- Please remove the accession numbers from the Results section and put them in the appropriate places in the methods section.
- Replace “Length of reference genome” column from Table 1 as this is identical with a column on the length of the individual de novo assemblies.
- Supplementary Figures are named “Figures”.
- Supplementary Figures and Tables should be numbered from one and not continuously from the Figures and Tables in the manuscript.
- Figure 1 legend: 43-13838 and 46-5069 have 14 SNPs in common according to the figure – not 7. Additionally please indicate colours and that it is a Venn diagram.
- Please be consistent in the methods section on whether “version” is named in parenthesis or without.


Spelling corrections:
- Line 60: “aligned to the reference genome”
- Line 131: Delete the affiliations so that it becomes “patients at the Mahidol Univeristy, Thailand”. The complete affiliation is listed in the methods.
- Line 134: “and compared to the reference strain”
- Line 161: “less than 45 bases were discarded using Trimmomatic” (replace dropped with discarded).
- Line 161: “The trimmed sequences were aligned to the reference strain”
- Line 162: “using the short read aligner Bowtie2” (add “the” and delete “sequence”)
- Line 164: “Variant calling was performed using the Genome Analysis Tool Kit” (remove s from Variants and uppercase letters for GATK)
- Line 170: “confirmed using Integrated Genomic Viewer” (uppercase IGV).
- Line 180: “Draft genome assembly” (non-capital a in assembly)
- Line 181: “The paired-end raw reads” (change paired to paired end)
- Line 183: “The algorithm works by using a de Bruijn graph”
- Line 268 and 273: Delete new-line at the end of the line
- Line 289: “within Thailand” (change with to within)
- Line 333: Write cycA in italics
- Line 408: “genotypes, countries” (add “s” to genotype)
- Line 410: “first draft genomes of M. tuberculosis Nonthaburi” (add “s” to genome)

·

Basic reporting

The aritcle is clear overall and the use of the genomic data for M. turbecolosis nonthaburi will be helpful to researchers trying to tacke TB around the world.

Experimental design

no comments

Validity of the findings

no comments

Additional comments

It would be nice to have a phylogenetic tree showing how Nonthaburi differs from other M. tuberculosis.

Reviewer 4 ·

Basic reporting

The article is almost of sufficient English quality. However, there are sections that appear hastily written and sloppy, which obscures the clarity of the text. If English is not the first language of the authors, the article could be read by a professional English editor, or by a peer or colleague. There are also several typographical errors that must be fixed.

The abstract could be more clearly written. Moreover, one of the highlights and novelties of the study is that not much WGS data for Mtb isolates from TB meningitis patients exists. This could be more clearly highlighted in the abstract.

The article doesn't conform to the required standards for the PeerJ journal. The journal requires that the results and discussion sections are written separately. This should be amended.

In both sections describing SNPS: 'Single nucleotide polymorphisms' and 'Polymorphisms in virulence genes, efflux pump related genes, ...' the reader would benefit from the data being reported in a table or figure format, together with support for SNPs in particular regions.

There are also several statements for which suitable references exist and must be provided:
-Line 106: The morbidity and mortality rate of TBM is higher than PTB.
-Line 155: Whole genome sequencing analysis - references of published work (where available) must be provided for all software used here and for de novo assembly.

The institutional affiliation for Taane G Clark is not clear.

Experimental design

The experimental design is generally good, and the samples are novel.

Major comment:
- It is not clear which WGS analysis techniques are used to infer SNPs that are suggested to occur in both resistance associated genes and in virulence associated/efflux genes. Was it the data from de novo assembly or was it the data from mapping the genome to the reference? This should be clarified and the potential caveats to each type of analysis should be mentioned. Moreover, given that both types of analysis were conducted, it would be useful to know whether the results for SNP calling are the same for both approaches?
- It is well known that short read WGS data is not reliable for detecting SNPs in repetitive regions such as PPE/PGRS genes, and in most cases these regions are filtered out of the analysis due to unreliable data. SNPs found in repetitive regions must be confirmed by PCR if they are to be reported in the manuscript. Or they must be removed from the report as they are unreliable.
- The reported fold coverage is incredibly high - higher than for any other Mtb WGS manuscript I have reviewed/read or any data I have worked with. The way this was calculated must be reported as it speaks to the reliability of the data.
- Given the major section on drug resistance, the authors could run the data through one of the recently developed (and publicly available) algorithms for detecting drug resistance in Mtb (Coll et al., 2015; Bradley et al., 2015) to see how the data compare.

Some minor comments:
- More information on the outcomes of disease for patients from which the Mtb isolates were obtained would be interesting and add to the data, especially given the discussion about drug susceptibility later on. Moreover, information about the HIV status of the patients would be interesting - especially given the discussion about host-interacting genes later on.
- The kind of stock and processing steps of the isolates (if known) should be briefly described. It is becoming well accepted that subculturing can impact WGS output and it is important to clarify here.
- Line 168: Filtering parameters for WGS are incredibly important for the resultant data. The "hard filter parameters" for GATK must be described.
- Line 171: The database from which gene function must be mentioned and cited.
- Clarity on when the phenotypic drug susceptibility was conducted must be provided - was it the initial isolate from the patient, or was it the stock that was sequenced? This could also impact the results of DS testing.

Validity of the findings

The authors have basically provided a description of polymorphisms in interesting genes/regions. While this is useful descriptive information, the manuscript could be made richer by a discussion of whether these mutations could have a specific role in causing TB meningitis specifically (given that this is the novelty of the sample set). Or how phenotypic relevance of the polymorphisms could be validated.

The findings are good and the data is valid for the most part. As described above, the only parts that require specific reworking are the report of SNPs in repetitive regions (given that Illumina short read sequencing is not reliable for these kinds of inferences). These SNPs should either be confirmed using PCR and Sanger sequencing or be removed from the manuscript. Moreover, the fold coverage should be elaborated on as it is extremely high. Finally, the data could be run through one of the tools for detection of DR in Mtb using WGS that have recently been made available. This would be more useful that providing a list of SNPs. These three points constitute the major revisions that in the opinion of this review are required for publication.

---

## Round 0.2 · accepted · Accept

Having addressed the comments of the reviewers satisfactorily, I have no further objections to the publication of the article.

Reviewer 1 ·

Basic reporting

The reading of the article has been improved which make it clear to understand. The introduction is comprehensible and well referenced.
Raw data is now public and the figures seems to conform PeerJ policy.

Experimental design

The contents of the article are new, interesting and well analysed. TB studies should be improved and these findings can contribute to better know the implication of the organism in the disease and in the resistance mechanisms.

Validity of the findings

The parameters used for all software sounds strictly enough to remove FP.
I'd have appreciated avoid using proprietary software, such as CCLC Bio, because algorithms and parameters are black boxes. Other more accepted softwares in the field (such as velvet, mira ) should have been taking into account.